# Replica Field Theory for a Generalized Franz–Parisi Potential of Inhomogeneous Glassy Systems: New Closure and the Associated Self-Consistent Equation

**DOI:** 10.3390/e26030241

**Published:** 2024-03-08

**Authors:** Hiroshi Frusawa

**Affiliations:** Laboratory of Statistical Physics, Kochi University of Technology, Tosa-Yamada, Kochi 782-8502, Japan; frusawa.hiroshi@kochi-tech.ac.jp

**Keywords:** supercooled liquids, replica theory, Franz–Parisi potential, density functional theory, self-consistent equation

## Abstract

On approaching the dynamical transition temperature, supercooled liquids show heterogeneity over space and time. Static replica theory investigates the dynamical crossover in terms of the free energy landscape (FEL). Two kinds of static approaches have provided a self-consistent equation for determining this crossover, similar to the mode coupling theory for glassy dynamics. One uses the Morita–Hiroike formalism of the liquid state theory, whereas the other relies on the density functional theory (DFT). Each of the two approaches has advantages in terms of perturbative field theory. Here, we develop a replica field theory that has the benefits from both formulations. We introduce the generalized Franz–Parisi potential to formulate a correlation functional. Considering fluctuations around an inhomogeneous density determined by the Ramakrishnan–Yussouf DFT, we find a new closure as the stability condition of the correlation functional. The closure leads to the self-consistent equation involving the triplet direct correlation function. The present field theory further helps us study the FEL beyond the mean-field approximation.

## 1. Introduction

Glass, an amorphous solid with elasticity, has a microscopic structure in which localized particles oscillate around their mean positions of a random lattice [1,2,3,4,5,6]. The spatial randomness is self-generated by the particle localization that breaks translational symmetry. A remarkable feature of the random structure is that glass microscopically lacks the long-range order and is similar to liquid in terms of density–density correlations [1,2,3,4,5,6]. As a precursor to the random structure of glass, supercooled liquids show heterogeneity over space and time [1,2,3,4,5,6,7,8,9,10,11]. The dynamical heterogeneity and facilitation [4,7,8,9,10,11,12,13,14] emerges on approaching the dynamical transition temperature ($T_{d}$), accompanied by the crossover from relaxational to activated dynamics. For $T>T_{d}$, transport is not collective on a large scale. For $T\to T_{d}$, the system gets stuck in a glassy metastable state, and the dynamical behaviors of supercooled liquids exhibit features such as a two-step decay with a first relaxation (β-relaxation) to a plateau followed by a stretched exponential relaxation (α-relaxation) of density fluctuations. Along with this dynamical crossover, the dynamics become progressively heterogeneous and correlated in space. For $T<T_{d}$, the relaxation times of the two-step decay increase rapidly despite slight changes in the disordered microstructure.

Various theories have tried to explain the dynamical heterogeneity and facilitation, as well as the dynamical crossover at $T_{d}$. These include either elasticity theory or kinetically constrained models focusing on the dynamical facilitation [4,12,13,14], and the mode coupling theory (MCT) [15,16], a dynamical theory relevant when approaching $T_{d}$ from the liquid phase. The MCT describes the onset of the two-step relaxation above $T_{d}$ and predicts the divergence of β-relaxation time at $T_{d}$. Extension to the inhomogeneous MCT further allows us to describe a growing dynamical heterogeneity using a time-dependent three-or four-point correlation function [2,7,8,9,10,11]. Yet the dynamical transition temperature $T_{d}$ is higher than the glass transition temperature observed in simulation and experimental studies. An interpretation of this discrepancy is that a mean-field description of the MCT is beyond the scope of the barrier-dominated dynamics between metastable states, though applicable to the relaxation dynamics within a metastable state. The divergent behavior of the two-step decay predicted by the MCT at $T_{d}$ becomes incomplete because of the activated events remaining in actual liquids for $T\leq T_{d}$ [15,16].

The activation dynamics dominant below $T_{d}$ are due to transitions between metastable states [15,16,17,18,19,20]. Therefore, the dynamical crossover implies the emergence of many metastable states at $T_{d}$ or the appearance of a free energy landscape (FEL) characterized by an exponentially large number of metastable states below $T_{d}$. From the thermodynamic point of view, we can describe the characteristic of the FEL using the configurational entropy obtained from the logarithm of the number of metastable states [1,2,17,18,19,20]. The Adam–Gibbs relation provides results in quantitative agreement with simulation and experimental results, relating the drastic changes in the relaxation time and viscosity to the decrease of the configurational entropy on approaching the glass transition temperature [17,18,19,20,21,22]. For example, simulation studies on mixtures interacting via the Lennard-Jones potential and its repulsive counterpart, the WCA one, demonstrate that these systems exhibit quite different dynamics despite having nearly identical structures [23,24,25,26,27,28]. Such a large difference in the dynamics is ascribable to a considerable gap between the configurational entropies while making a slight difference between the two-point correlation functions. Previous investigations confirmed that the configurational entropies associated with correlation functions differ greatly between the Lennard-Jones and WCA mixtures despite the structural similarity, therefore predicting the distinct dynamical behaviors from the Adam–Gibbs relation [23,24,25,26,27,28].

Static approaches, other than dynamical ones such as the MCT, are relevant to investigate the FEL or the configurational entropy [17,18,19,20,29,30,31,32,33,34,35,36,37,38,39,40,41,42,43,44,45,46,47,48,49,50,51,52,53,54,55,56,57]. These include replica theory [17,18,19,20,29,30,31,32,33,34,35,36,37,38], density functional theory (DFT) [39,40,41,42,43,44,45,46,47,48,49,50,51,52], and a combination of the replica theory and DFT [53,54,55,56,57]. The static theories commonly focus on local minima of free-energy functionals without considering fluctuations due to the mean-field approximation. On the one hand, the DFT determines the metastable state by exploring a local minimum of the free-energy density functional [39,40,41,42,43,44,45,46,47,48,49,50,51,52]. Given the inhomogeneous density distribution as overlapping Gaussians centered around a random lattice, previous studies have confirmed that Gaussian distribution with a large spread creates the optimum density profile. The low degree of localization around the random lattice is consistent with experimental and simulation results. On the other hand, replica theory considers a system of coupled *m*-replicas of the original system [17,18,19,20,29,30,31,32,33,34,35,36,37,38,53,54,55,56,57]. The replica free-energy functional depends on a two-point correlation function between two copies (an inter-replica correlation function), an order parameter measuring the degree of similarity between two typical configurations. We obtain the correct result by taking the limit of m→1 with the inter-replica coupling switched off. While the order parameter goes to zero in the liquid state without the inter-replica coupling, the order parameter in an ergodicity-broken phase has a finite value because two copies remain highly correlated even after switching off the inter-replica coupling. The replica theory has successfully explained experimental and simulation results using the following four approximations: the small cage expansion [17,18,19,20,31,32], the effective potential approximation [17,18,19,20,31,32], the replicated hypernetted-chain (RHNC) approximation [33,34,35,36,37,38], and the third-order functional expansion in DFT [56,57,58,59,60,61]. While the first two are perturbation methods with the local cage size as a reference scale, the last two approximations cover those of the liquid-state theory [62,63,64].

The Franz–Parisi (FP) potential obtained in the RHNC approximation serves as a starting point for this paper. The FP potential [65,66,67,68,69,70,71,72,73,74,75] is a function of overlap *Q*, a weighted average over the system of the two-point correlation function, and plays the same role as the Landau free energy of a global parameter *Q* that indicates a distance between the two copies in configuration space. Theoretical and simulation studies have demonstrated that the FP potential reproduces the temperature evolution of FELs, just like the Landau free energy [65,66,67,68,69,70,71,72,73,74,75]. With decreasing temperature, the FP potential develops a secondary minimum for Q>0 representing a metastable state. Considering Q=0 in the liquid state, we can see that the potential difference, V(Q)−V(0), corresponds to the entropic cost of localizing the system in a single metastable state (i.e., the configurational entropy).

In this paper, we generalize the FP potential by fixing an inter-replica correlation function instead of the overlap *Q*. We formulate the generalized FP potential by developing a new framework that is beneficial to investigate the FEL while considering inhomogeneous supercooled liquids with the help of field theoretical method. A field theory combining the DFT [53,54,55,56,57,58,59,60,61] and replica theory [17,18,19,20,29,30,31,32,33,34,35,36,37,38,53,54,55,56,57] forms the basis of our framework. There are two requirements to be satisfied by the field theory and the associated functional. The first requirement is that the developed framework can consider inhomogeneous systems. The second requirement is that the generalized FP functional applies to non-equilibrium states away from metastable states. To meet the requirements, this paper presents the correlation functional theory that provides the generalized FP potential functional without going through the Morita-Hiroike functional [33,34,35,36,37,38,62,63,64]. The generalized FP potential has three features as a functional of density and correlation function. First, this potential is a functional of metastable density that becomes equal to that of the DFT in the limit of m→1. Second, the field-theoretical perturbation method allows us to have a new correlation functional different from the Morita–Hiroike one while maintaining consistency with the liquid theory in that the approximate form reduces to the RHNC functional. Last, the potential functional of a given inter-replica correlation function has a minimum where a new closure reducible to the RHNC approximation [33,34,35,36,37,38,62,63] holds. A remarkable result is that an approximation of the new closure yields the self-consistent equation for a non-ergodicity parameter that includes the triplet direct correlation function (DCF) [62,63,76,77,78], similar to that formulated by either the MCT [15,16] or the replica theory [37,56,57], respectively.

The paper is organized as follows. In Section 2, we define the generalized FP potential. Comparison between the generalized and original FP potentials clarifies what we modify through the generalization. Section 3 summarizes the theoretical results consisting of four parts as follows: relation for obtaining the generalized FP potential from the grand potential of *m*-replica system with inter-replica correlation function fixed (*Result 1*); functional form of the constrained grand potential (*Result 2*); new closure for two-point correlation function (*Result 3*); the associated self-consistent equation for a non-ergodicity parameter (*Result 4*). We obtain the generalized FP potential from *Result 2* with the help of the relation in *Result 1*. The extremum condition of this potential yields a new closure in *Result 3*. It also turns out that a self-consistent equation obtained from an approximate form of the closure involves the triplet DCF as presented in *Result 4*. In Section 4, we calculate the perturbative terms using a strong-coupling perturbation theory developed for obtaining *Result 2* (see Appendix B). In the saddle-point approximation, the strong-coupling perturbation theory provides the correlation functional form of the constrained grand potential given in *Result 2*. In Section 5, we make some concluding remarks.

## 2. Generalized Franz–Parisi (FP) Potential

We generalize the FP potential in comparison with its original definition.

### 2.1. The Original FP Potential

Let $C_{a}$ be a configuration that represents a set of *N*-particle positions, ${\left\{r_{a,i}\right\}}_{i=1,\cdots ,N}$, in replica *a* (1≤a≤m) when considering *m* copies of the liquid. The overlap $\hat{Q}\left({\hat{\rho }}_{a},{\hat{\rho }}_{b}\right)$ (a≠b) measures the degree of similarity between a pair of replicas using the microscopic density (or the so-called density “operator” [79,80,81,82,83,84,85,86,87,88]) in replica *a*, ${\hat{\rho }}_{a}^{\left(N\right)}\left(r\right)=\sum_{i=1}^{N}\delta \left(r-r_{a,i}\right)$. We define that
(1)$$\begin{array}{c} \hat{Q}\left({\hat{\rho }}_{a},{\hat{\rho }}_{b}\right)=\frac{1}{N}\int \;\;\;\int dr\;dr^{\prime }{\hat{\rho }}_{a}^{\left(N\right)}\left(r\right){\hat{\rho }}_{b}^{\left(N\right)}\left(r^{\prime }\right)\eta \left(r-r^{\prime }\right), \end{array}$$
where a distribution function η(r) specifies the spatial averaging performed over a finite range; for example, we have η(r)=Θ(a−|r|) using the Heaviside function Θ(r) and particle diameter *a* [65,66,67,68,69,70,71,72,73,74,75].

The FP potential V(Q) is obtained in two steps [65,66,67,68,69,70,71,72,73,74,75]. First, we fix a reference configuration ${\hat{\rho }}_{1}$ of replica 1, which plays the role of quenched variable in the effective potential $V(Q,{\hat{\rho }}_{1})$ as seen from the following definition: (2)$$e^{-\beta NV^{+}\left(Q,{\hat{\rho }}_{1}\right)}\;=\sum_{C_{a}}\;e^{-\beta U_{a}\left({\hat{\rho }}_{1},{\hat{\rho }}_{a}\right)}\delta \left[\;Q-\hat{Q}\left({\hat{\rho }}_{1},{\hat{\rho }}_{a}\right)\right],$$
(3)$$V\left(Q,{\hat{\rho }}_{1}\right)=\lim_{U_{\mathrm{inter}}\to 0}V^{+}\left(Q,{\hat{\rho }}_{1}\right),$$
where $\sum_{C_{a}}$ denotes $\left(1/N!\right)\int \cdots \int dr_{a,1}\cdots dr_{a,N}$ in the canonical ensemble of replica *a* for a≥2, β the inverse thermal energy ${\left(k_{B}T\right)}^{-1}$, and $U_{a}\left({\hat{\rho }}_{1},{\hat{\rho }}_{a}\right)$ the interaction energy of replica *a* that is the sum of intra-replica interaction energy $U_{\mathrm{intra}}\left({\hat{\rho }}_{a}\right)$ and inter-replica one $U_{\mathrm{inter}}\left({\hat{\rho }}_{1},{\hat{\rho }}_{a}\right)$:(4)$$U_{a}\left({\hat{\rho }}_{1},{\hat{\rho }}_{a}\right)=U_{\mathrm{intra}}\left({\hat{\rho }}_{a}\right)+U_{\mathrm{inter}}\left({\hat{\rho }}_{1},{\hat{\rho }}_{a}\right),$$
where
(5)$$\;U_{\mathrm{intra}}\left({\hat{\rho }}_{a}\right)=\frac{1}{2}\int \;\;\;\int drdr^{\prime }\left\{{\hat{\rho }}_{a}^{\left(N\right)}\left(r\right)v\left(r-r^{\prime }\right){\hat{\rho }}_{a}^{\left(N\right)}\left(r^{\prime }\right)-{\hat{\rho }}_{a}^{\left(N\right)}\left(r\right)v\left(r-r^{\prime }\right)\delta \left(r-r^{\prime }\right)\right\},$$
(6)$$\;U_{\mathrm{inter}}\left({\hat{\rho }}_{1},{\hat{\rho }}_{a}\right)=\int \;\;\;\int drdr^{\prime }{\hat{\rho }}_{1}^{\left(N\right)}\left(r\right)\tilde{v}\left(r-r^{\prime }\right){\hat{\rho }}_{a}^{\left(N\right)}\left(r^{\prime }\right),$$
using the intra-replica interaction potential v(r) and the inter-replica one $\tilde{v}\left(r\right)$. It is noted that the effective potential $V(Q,{\hat{\rho }}_{1})$ is defined in the absence of inter-replica interactions as represented by Equation (Equation 3).

Next, we perform the canonical average of $V(Q,{\hat{\rho }}_{1})$ over all possible choices for the reference configuration with the statistical weight $p_{\mathrm{eq}}\left({\hat{\rho }}_{1}\right)$ as follows: (7)$$V\left(Q\right)=\sum_{C_{1}}\;p_{\mathrm{eq}}\left({\hat{\rho }}_{1}\right)\;V\left(Q,{\hat{\rho }}_{1}\right),$$
(8)$$p_{\mathrm{eq}}\left({\hat{\rho }}_{1}\right)=\frac{e^{-\beta U_{\mathrm{intra}}\left({\hat{\rho }}_{1}\right)}}{\sum_{C_{1}}e^{-\beta U_{\mathrm{intra}}\left({\hat{\rho }}_{1}\right)}}.$$
The replica trick allows us to calculate Equation (Equation 7), thus obtaining the FP potential V(Q) of the Landau type.

### 2.2. Generalization

Here, we introduce a generalized FP potential $W(\tilde{G})$ as a functional of prescribed correlation function $\tilde{G}\left(r,r^{\prime }\right)$, instead of the overlap *Q*. In terms of the Landau theory, we consider a local order parameter, instead of the global one. We use the grand canonical ensemble represented by the following operator:(9)$$\begin{array}{c} {\mathrm{Tr}}_{a}\equiv \sum_{N=0}^{\infty }\frac{e^{N\beta \mu }}{N!}\;\int dr_{a,1}\cdots \int dr_{a,N}=\sum_{N=0}^{\infty }e^{N\beta \mu }\sum_{C_{a}}, \end{array}$$
where the chemical potential βμ in units of $k_{B}T$ determines the most probable number $N^{*}$, thereby providing the uniform density $\bar{\rho }=N^{*}/V$ common to each replica with volume *V*.

Given a reference configuration $C_{1}$ of replica 1, we have the interaction energy $U_{a}\left({\hat{\rho }}_{1},{\hat{\rho }}_{a}\right)$ of replica *a*, providing the grand potential $\omega_{a}\left({\hat{\rho }}_{1}\right)$ of replica *a* as follows:(10)$$\begin{array}{ll} e^{-\beta \omega_{a}^{+}\left({\hat{\rho }}_{1}\right)} & ={\mathrm{Tr}}_{a}\;e^{-\beta U_{a}\left({\hat{\rho }}_{1},{\hat{\rho }}_{a}\right)}\\ \; & =\int D\tilde{G}\;{\mathrm{Tr}}_{a}\;e^{-\beta U_{a}\left({\hat{\rho }}_{1},{\hat{\rho }}_{a}\right)}\prod_{b=1,a}I_{b}\left(\rho ,\hat{\rho }\right)\;\Delta_{a}\left(\tilde{G},\rho \right)\\ \; & =\int D\tilde{G}\;e^{-\beta N^{*}W\left(\tilde{G},{\hat{\rho }}_{1}\right)}, \end{array}$$
(11)$$\omega_{a}\left({\hat{\rho }}_{1}\right)=\lim_{U_{\mathrm{inter}}\to 0}\omega_{a}^{+}\left({\hat{\rho }}_{1}\right),$$
where the functional integral representation in Equation (Equation 10) is obtained from multiplying the right-hand side (rhs) of the first line in Equation (Equation 10) by the following identity:(12)$$\begin{array}{cc} 1 & =\int D\tilde{G}\prod_{b=1,a}\int D\rho_{b}\;\prod_{\left\{r\right\},\left\{r^{\prime }\right\}}\prod_{\left\{r\right\}}\delta \left[\rho_{b}\left(r\right)-{\hat{\rho }}_{b}^{\left(N\right)}\left(r\right)\right]\;\delta \left[\tilde{G}\left(r,r^{\prime }\right)-\rho_{1}\left(r\right)\rho_{a}\left(r^{\prime }\right)\right]\\ & =\int D\tilde{G}\;\prod_{b=1,a}I_{b}\left(\rho ,\hat{\rho }\right)\;\Delta_{a}\left(\tilde{G},\rho \right). \end{array}$$
Equation (Equation 12) implies that
(13)$$I_{b}\left(\rho ,\hat{\rho }\right)\equiv \int D\rho_{b}\prod_{\left\{r\right\}}\delta \left[\rho_{b}\left(r\right)-{\hat{\rho }}_{b}^{\left(N\right)}\left(r\right)\right]=1,$$
(14)$$\Delta_{a}\left(\tilde{G},\rho \right)\equiv \prod_{\left\{r\right\},\left\{r^{\prime }\right\}}\delta \left[\tilde{G}\left(r,r^{\prime }\right)-\rho_{1}\left(r\right)\rho_{a}\left(r^{\prime }\right)\right].$$
The relation (Equation 13) at b=1 represents that only the density distribution ${\hat{\rho }}_{1}^{\left(N\right)}\left(r\right)$ is allowed due to a fixed configuration $C_{1}$ of replica 1.

Equations (Equation 10)–(Equation 14) reveal that the field-theoretical formulation of the effective potential $W(\tilde{G},{\hat{\rho }}_{1})$ can be developed as follows [79,80,81,82,83,84,85,86,87,88]:(15)$$\begin{array}{cc} W(\tilde{G},{\hat{\rho }}_{1}) & =\lim_{U_{\mathrm{inter}}\to 0}W^{+}\left(\tilde{G},{\hat{\rho }}_{1}\right), \end{array}$$
where
(16)$$\begin{array}{cc} e^{-\beta N^{*}W^{+}\left(\tilde{G},{\hat{\rho }}_{1}\right)} & ={\mathrm{Tr}}_{a}\;e^{-\beta U_{a}\left({\hat{\rho }}_{1},{\hat{\rho }}_{a}\right)}\prod_{b=1,a}I_{b}\left(\rho ,\hat{\rho }\right)\;\Delta_{a}\left(\tilde{G},\rho \right)\\ & =I_{1}\left(\rho ,\hat{\rho }\right)\;{\mathrm{Tr}}_{a}\;I_{a}\left(\rho ,\hat{\rho }\right)\;e^{-\beta U_{a}\left(\rho_{1},\rho_{a}\right)}\;\Delta_{a}\left(\tilde{G},\rho \right)\\ & =I_{1}\left(\rho ,\hat{\rho }\right)\int D\rho_{a}\;e^{-\beta U_{a}\left(\rho_{1},\rho_{a}\right)}\;{\mathrm{Tr}}_{a}\;\prod_{\left\{r\right\}}\delta \left[\rho_{a}\left(r\right)-{\hat{\rho }}_{a}^{\left(N\right)}\left(r\right)\right]\;\Delta_{a}\left(\tilde{G},\rho \right)\\ & =I_{1}\left(\rho ,\hat{\rho }\right)\int D\rho_{a}\;e^{-\beta \left\{U_{a}\left(\rho_{1},\rho_{a}\right)-TS_{a}^{\mathrm{id}}\left(\rho_{a}\right)\right\}}\;\Delta_{a}\left(\tilde{G},\rho \right). \end{array}$$
In the last line of Equation (Equation 16), we have the ideal gas entropy defined by
(17)$$\begin{array}{c} -TS_{a}^{\mathrm{id}}\left(\rho_{a}\right)=k_{B}T\int dr\;\rho_{a}\left(r\right)\left\{\ln\rho_{a}\left(r\right)-1-\beta \mu \right\}. \end{array}$$
The generalized FP potential $W(\tilde{G})$ is obtained from the grand canonical average of $W(\tilde{G},{\hat{\rho }}_{1})$ for the reference configuration as follows: (18)$$W\left(\tilde{G}\right)={\mathrm{Tr}}_{1}\;P_{\mathrm{eq}}\left({\hat{\rho }}_{1}\right)W\left(\tilde{G},{\hat{\rho }}_{1}\right),$$
(19)$$P_{\mathrm{eq}}\left({\hat{\rho }}_{1}\right)=\frac{e^{-\beta U_{\mathrm{intra}}\left({\hat{\rho }}_{1}\right)}}{{\mathrm{Tr}}_{1}\;e^{-\beta U_{\mathrm{intra}}\left({\hat{\rho }}_{1}\right)}},$$
similar to Equations (Equation 7) and (Equation 8). Equation (Equation 18) clarifies that a given configuration ${\hat{\rho }}_{1}$ plays a role of quenched disorder to another replica *a* [65,66,67,68,69,70,71,72,73,74,75]. Since we consider all possible configurations of ${\hat{\rho }}_{1}$, the statistical weight $P_{\mathrm{eq}}\left({\hat{\rho }}_{1}\right)$ is of the Boltzmann form as well as $p_{\mathrm{eq}}\left({\hat{\rho }}_{1}\right)$ in Equation (Equation 8).

Several remarks on Equations (Equation 13)–(19) are in order:Equation (Equation 14) tells us that a prescribed correlation field $\tilde{G}\left(r,r^{\prime }\right)$ represents a product $\rho_{1}\left(r\right)\rho_{a}\left(r^{\prime }\right)$ of two instantaneous density distributions in different replicas, or a statistical realization of density-density correlation function [62,63].To perform the configurational integral ${\mathrm{Tr}}_{a}$ in the second line on the rhs of Equation (Equation 16), it is indispensable to introduce the Fourier transform representation of the delta functional using the functional integral over the one-body potential field, which is dual to the density field $\rho_{a}\left(r\right)$ [84,85,86,87,88]. The ideal gas entropy given by Equation (Equation 17) appears in the last line of Equation (Equation 16) due to the saddle-point approximation of the one-body potential field [84,85,86,87,88].When different replica particles form complexes because of the attractive inter-replica interactions between them (i.e., $\tilde{v}\left(r\right)<0$), we have $\tilde{G}\left(r\right)/{\bar{\rho }}^{2}\gg 1$ in an overlapped region (e.g., |r|≤a), thereby providing a significant value of overlap *Q* that is greater than the random overlap obtained from $\tilde{G}\left(r\right)={\bar{\rho }}^{2}$. The glassy state preserves an overlapped state due to frozen configurations of particles even after the attractive inter-replica interactions are switched off (i.e., $\tilde{v}\left(r\right)\to 0$). The generalized FP potential $W(\tilde{G})$ is available to explore such an overlapped state that is locally stable.It is also noted that the above formalism presented in Equations (Equation 10)–(Equation 17) has been conventionally used for the formulation of continuous field theory [79,80,81,82,83,84,85,86,87,88]; the density operator ${\hat{\rho }}_{b}^{\left(N\right)}\left(r\right)$ (b=1,a) has been mapped to a density field $\rho_{b}\left(r\right)$ using the density functional integral in Equations (Equation 13) and (Equation 16) according to the conventional formalism in statistical field theory [79] (see also the literature [80,81,82,83,84,85,86,87,88,89,90,91] for discussions about the underlying physics of this formal procedure to introduce a continuous density field).

## 3. Main Results

We present four sets of main results based on the strong-coupling perturbation theory (see Appendix B for details). Figure 1 summarizes the results schematically.

### 3.1. *Result 1*: Replica Formalism of the Generalized FP Potential

Let $\Omega_{v}^{\left(m\right)}\left(\tilde{G}\right)$ be the constrained grand potential of *m* replicas defined by
(20)$$\begin{array}{cc} e^{-\beta \Omega_{v}^{\left(m\right)}\left(\tilde{G}\right)} & =\mathrm{Tr}\;e^{-\beta U(v,\hat{\rho })}\;\prod_{a=2}^{m}\Delta_{a}\left(\tilde{G},\hat{\rho }\right)\\ & =\int D\rho \;\mathrm{Tr}\;e^{-\beta U(v,\rho )}\prod_{b=1}^{m}\prod_{\left\{r\right\}}\delta \left[\rho_{b}\left(r\right)-{\hat{\rho }}_{b}^{\left(N\right)}\left(r\right)\right]\;\prod_{a=2}^{m}\Delta_{a}\left(\tilde{G},\rho \right) \end{array}$$
where $\int D\rho \equiv \prod_{b=1}^{m}\int D\rho_{b}$, $\mathrm{Tr}\equiv \prod_{b=1}^{m}{\mathrm{Tr}}_{b}$, the matrix elements of v are $v_{ab}\left(r\right)=0$ (a≠b) and $v_{aa}\left(r\right)=v\left(r\right)$, $\hat{\rho }={\left({\hat{\rho }}_{1}^{\left(N\right)},\cdots ,{\hat{\rho }}_{m}^{\left(N\right)}\right)}^{T}$, $\rho ={\left(\rho_{1},\cdots ,\rho_{m}\right)}^{T}$, and the interaction energy $U(v,\hat{\rho })$ in Equation (Equation 20) is given by
(21)$$\begin{array}{c} U\left(v,\hat{\rho }\right)=\frac{1}{2}\int \;\;\;\int drdr^{\prime }\left\{\hat{\rho }{\left(r\right)}^{T}v\left(r-r^{\prime }\right)\hat{\rho }\left(r^{\prime }\right)-\sum_{b=1}^{m}{\hat{\rho }}_{b}^{\left(N\right)}\left(r\right)v\left(r-r^{\prime }\right)\delta \left(r-r^{\prime }\right)\right\}, \end{array}$$
excluding the intra-replica self-energy. Incidentally, there are two methods to treat the density functional integral in Equation (Equation 20) [79,80,81,82,83,84,85,86,87,88], both of which will be utilized as seen from Equations (Equation 86) and (Equation 108).

It is readily seen from Equations (Equation 16) and (Equation 20) that the constrained grand potential $\Omega_{v}^{\left(m\right)}\left(\tilde{G}\right)$ is expressed using $W(\tilde{G},{\hat{\rho }}_{1})$ as
(22)$$\begin{array}{c} e^{-\beta \Omega_{v}^{\left(m\right)}\left(\tilde{G}\right)}={\mathrm{Tr}}_{1}\;e^{-\beta U_{\mathrm{intra}}\left({\hat{\rho }}_{1}\right)-\left(m-1\right)\beta N^{*}W\left(\tilde{G},{\hat{\rho }}_{1}\right)}. \end{array}$$
The replica trick allows us to have the relation between the constrained grand potential $\Omega_{v}^{\left(m\right)}$ and the generalized FP potential $W(\tilde{G})$:(23)$$\begin{array}{cc} N^{*}W\left(\tilde{G}\right) & =\lim_{m\to 1}\frac{\partial \Omega_{v}^{\left(m\right)}\left(\tilde{G}\right)}{\partial m}, \end{array}$$
which is the first result (*Result 1*; see Appendix A for the detailed derivation). It is noted that the conventional replica trick proves the necessity of m→1 to consider the quenched type of the FP formalism, though it has been physically motivated to take the limit of m→1 based on the Monasson formalism [18,22,75].

### 3.2. *Result 2*: The Constrained Grand Potential Functional of *m* Replicas in an Inhomogeneous State

In *Result 2*, we provide the correlation functional form of the constrained grand potential $\Omega_{v}^{\left(m\right)}\left(\tilde{G}\right)$. Section 4 will sketch how the perturbative field theory developed in Appendix B yields the correlation functional given in *Result 2*.

Let us consider the inhomogeneous system characterized by the mean-field density $\rho_{a}^{*}\left(r\right)$ satisfying
(24)$$\begin{array}{c} \rho_{a}^{*}\left(r\right)=e^{\beta \mu -\frac{c_{aa}\left(0\right)}{2}}\exp\left\{\sum_{b=1}^{m}\int dr^{\prime }c_{ab}\left(r-r^{\prime }\right)\rho_{b}^{*}\left(r^{\prime }\right)\right\}, \end{array}$$
where $c_{ab}\left(r\right)$ denotes the two-point DCF (simply called DCF) between replica *a* and replica *b*. Here we suppose that a given function $\tilde{G}\left(r,r^{\prime }\right)$ imposed on the inter-replica correlation between replica 1 and replica *a* (a≥2) is expressed as
(25)$$\begin{array}{c} \tilde{G}\left(r,r^{\prime }\right)=\rho_{1}^{*}\left(r\right)\rho_{a}^{*}\left(r^{\prime }\right)\;\tilde{g}\left(r-r^{\prime }\right)=\rho_{1}^{*}\left(r\right)\rho_{a}^{*}\left(r^{\prime }\right)\;\left\{1+\tilde{h}\left(r-r^{\prime }\right)\right\}, \end{array}$$
using a statistical realization of inter-replica radial distribution function $\tilde{g}\left(r\right)$ or inter-replica total correlation function (TCF) $\tilde{h}\left(r\right)\equiv \tilde{g}\left(r\right)-1$ [62,63]. Namely, Equations (Equation 14) and (Equation 25) imply the constraint,
(26)$$\begin{array}{c} \rho_{1}^{*}\left(r\right)\rho_{a}^{*}\left(r^{\prime }\right)\left\{1+\tilde{h}\left(r-r^{\prime }\right)\right\}=\rho_{1}\left(r\right)\rho_{a}\left(r^{\prime }\right), \end{array}$$
on $\rho_{1}\left(r\right)\rho_{a}\left(r^{\prime }\right)$ which is a statistical realization of density-density correlation [62,63] as mentioned above. Equation (Equation 26) includes the trivial inter-replica constraints as follows: one constraint, ${\bar{\rho }}^{2}=\rho_{1}\left(r\right)\rho_{a}\left(r^{\prime }\right)$ (i.e., $\tilde{h}\left(r-r^{\prime }\right)=0$), forces the two-replica system to maintain uniformity without inter-replica correlations, whereas another constraint, $0=\rho_{1}\left(r\right)\rho_{a}\left(r^{\prime }\right)$ (i.e., $\tilde{h}\left(r-r^{\prime }\right)=-1$), imposes a region where two particles of different replicas exclude each other. In Section 3.3, we will see that the metastable TCF ${\tilde{h}}_{*}\left(r-r^{\prime }\right)$ corresponds to the TCF obtained from averaging over statistical realizations of instantaneous density-density correlation $\rho_{1}\left(r\right)\rho_{a}\left(r^{\prime }\right)$ consistently with Equation (Equation 26) as well as the liquid-state theory [62,63].

Let h(r) and c(r) be the correlation matrices of TCFs and DCFs, respectively. The intra-and inter-replica matrix elements vary, depending on whether replica 1 is included or not: when setting χ(r)=h(r) or c(r) with the subscripts of their matrix elements denoting a pair of replicas, $\chi_{11}\left(r\right)=\chi_{1}\left(r\right)$ and $\chi_{aa}\left(r\right)=\chi \left(r\right)$ for a≥2, whereas $\chi_{1a}\left(r\right)=\chi_{a1}\left(r\right)=\tilde{\chi }\left(r\right)$ for a≥2 and $\chi_{ab}\left(r\right)={\tilde{\chi }}^{\prime }\left(r\right)$ for a≠b and a,b≥2. As a consequence, we see from Equation (Equation 24) that
(27)$$\begin{array}{c} \rho_{1}^{*}\left(r\right)=e^{\beta \mu -\frac{c_{1}\left(0\right)}{2}}\exp\left\{\int dr^{\prime }c_{1}\left(r-r^{\prime }\right)\rho_{1}^{*}\left(r^{\prime }\right)+\left(m-1\right)\int dr^{\prime }\tilde{c}\left(r-r^{\prime }\right)\rho^{*}\left(r^{\prime }\right)\right\}, \end{array}$$
where $\rho_{a}^{*}\left(r\right)=\rho^{*}\left(r\right)$ for a≥2. It is noted that the metastable density distribution $\rho_{1}^{*}\left(r\right)$ reduces to that from the Ramakrishnan-Yussouf density functional [58,59,60,61]:(28)$$\begin{array}{cc} \rho_{1}^{*}\left(r\right) & =e^{\beta \mu -\frac{c_{1}\left(0\right)}{2}}\exp\left\{\int dr^{\prime }c_{1}\left(r-r^{\prime }\right)\rho_{1}^{*}\left(r^{\prime }\right)\right\}, \end{array}$$
in the limit of m→1.

The variational approach presented in Section B.1 justifies the following set of inhomogeneous Ornstein-Zernike equations [33,34,35,36,37,38,62,63]: in general, we have
(29)$$\begin{array}{c} h_{ac}\left(r-r^{\prime }\right)=c_{ac}\left(r-r^{\prime }\right)+\sum_{b=1}^{m}\int dr^{\prime }\rho_{b}^{*}\left(r\prime \prime \right)c_{ab}\left(r-r\prime \prime \right)h_{bc}\left(r\prime \prime -r^{\prime }\right), \end{array}$$
which reads
(30)$$\begin{array}{c} h_{1}\left(r-r^{\prime }\right)=c_{1}\left(r-r^{\prime }\right)+\int dr\prime \prime \rho_{1}^{*}\left(r\prime \prime \right)c_{1}\left(r-r\prime \prime \right)h_{1}\left(r\prime \prime -r^{\prime }\right)\\ +\left(m-1\right)\int dr\prime \prime \rho^{*}\left(r\prime \prime \right)\tilde{c}\left(r-r\prime \prime \right)\tilde{h}\left(r\prime \prime -r^{\prime }\right), \end{array}$$
and
(31)$$\begin{array}{cc} \tilde{h}\left(r-r^{\prime }\right)=\tilde{c}\left(r-r^{\prime }\right) & +\int dr\prime \prime \left\{\rho^{*}\left(r\prime \prime \right)\tilde{c}\left(r-r\prime \prime \right)h\left(r\prime \prime -r^{\prime }\right)+\rho_{1}^{*}\left(r\prime \prime \right)c_{1}\left(r-r\prime \prime \right)\tilde{h}\left(r\prime \prime -r^{\prime }\right)\right\}\\ & +\left(m-2\right)\int dr\prime \prime \rho^{*}\left(r\prime \prime \right)\tilde{c}\left(r-r\prime \prime \right)\tilde{h}\left(r\prime \prime -r^{\prime }\right), \end{array}$$
in agreement with previous expressions [33,34,35,36,37,38].

The second result (*Result 2*) can be obtained using the perturbative field theory at strong coupling (see Appendix B). It will be shown in Section 4 that the constrained grand potential is of the following functional form: (32)$$\begin{array}{cc} & \beta \Omega_{v^{\prime }}^{\left(m\right)}\left(\tilde{G}\right)\\ & \;=\frac{1}{2}\int \;\;\;\int dr_{0}dr\;\left\{\rho_{1}^{*}\left(r_{0}\right)\rho_{1}^{*}\left(r_{0}-r\right)+\left(m-1\right)\rho^{*}\left(r_{0}\right)\rho^{*}\left(r_{0}-r\right)\right\}g\left(r\right)v\left(r\right)\\ & \;+\left(m-1\right)\int \;\;\;\int dr_{0}dr\;\rho_{1}^{*}\left(r_{0}\right)\rho^{*}\left(r_{0}-r\right)\tilde{g}\left(r\right)\tilde{v}\left(r\right)\\ & \;+\int dr_{0}\;\left[\;\rho_{1}^{*}\left(r_{0}\right)\left\{\ln\rho_{1}^{*}\left(r_{0}\right)-1-\beta \mu \right\}+\left(m-1\right)\rho^{*}\left(r_{0}\right)\left\{\ln\rho^{*}\left(r_{0}\right)-1-\beta \mu \right\}\right]\\ & \;+\frac{1}{2}\int \;\;\;\int dr_{0}dr\;\left\{\rho_{1}^{*}\left(r_{0}\right)h_{1}\left(r\right)\delta \left(r\right)+\left(m-1\right)\rho^{*}\left(r_{0}\right)h\left(r\right)\delta \left(r\right)-\ln\left|S\right|\right\}\\ & \;+\left(m-1\right)\int \;\;\;\int dr_{0}dr\;\rho_{1}^{*}\left(r_{0}\right)\rho^{*}\left(r_{0}-r\right)\left\{\tilde{g}\left(r\right)\ln\tilde{g}\left(r\right)-\tilde{h}\left(r\right)-{\tilde{h}}^{2}\left(r\right)+e^{\tilde{h}\left(r\right)}-\tilde{g}\left(r\right)\right\}, \end{array}$$
where the matrix elements of $v^{\prime }$ has a non-zero potential $v_{1a}\left(r\right)=v_{a1}\left(r\right)=\tilde{v}\left(r\right)$ between replica 1 and replica *a* that enforces Equation (Equation 26) without the constraint $\Delta_{a}\left(\tilde{G},\hat{\rho }\right)$, and the matrix element of S is given by $S_{ab}\left(r\right)=\delta_{ab}\delta \left(r\right)+\rho_{a}^{*}\left(r_{0}\right)h_{ab}\left(r\right)$. It is noted that the last line of Equation (Equation 32) is reduced to the RHNC functional of $\tilde{h}\left(r\right)$ in the approximation of $e^{\tilde{h}\left(r\right)}-\tilde{g}\left(r\right)\approx {\tilde{h}}^{2}\left(r\right)/2$ [33,34,35,36,37,38,62,63,64,85,92].

### 3.3. *Result 3*: New Closure Obtained from the Generalized FP Potential

The stationary condition of $W(\tilde{G})$ given by Equation (Equation 23) can be written as
(33)$$\begin{array}{c} {\left.\frac{\delta W(\tilde{G})}{\delta \tilde{h}}\right|}_{\tilde{h}={\tilde{h}}_{*}}=\frac{1}{N^{*}}{\left.\lim_{\tilde{v}\to 0}\frac{\delta }{\delta \tilde{h}}\left\{\lim_{m\to 1}\frac{\partial \Omega_{v^{\prime }}^{\left(m\right)}\left(\tilde{G}\right)}{\partial m}\right\}\right|}_{\tilde{h}={\tilde{h}}_{*}}=0. \end{array}$$
It is found from Equation (Equation 32) that
(34)$$\begin{array}{cc} \lim_{m\to 1}\frac{\partial \beta \Omega_{v^{\prime }}^{\left(m\right)}\left(\tilde{G}\right)}{\partial m} & =\int \;\;\;\int dr_{0}dr\;\left\{\frac{1}{2}\rho^{*}\left(r_{0}\right)\rho^{*}\left(r_{0}-r\right)g\left(r\right)v\left(r\right)+\rho_{1}^{*}\left(r_{0}\right)\rho^{*}\left(r_{0}-r\right)\tilde{g}\left(r\right)\tilde{v}\left(r\right)\right\}\\ & +\int dr_{0}\;\rho^{*}\left(r_{0}\right)\left\{\ln\rho^{*}\left(r_{0}\right)-1-\beta \mu \right\}+\frac{1}{2}\int \;\;\;\int dr_{0}dr\;\rho^{*}\left(r_{0}\right)h\left(r\right)\delta \left(r\right)\\ & +\frac{1}{2}\int \;\;\;\int dr_{0}dr\;\rho_{1}^{*}\left(r_{0}\right)\rho^{*}\left(r_{0}-r\right)\tilde{c}\left(r\right)\tilde{h}\left(r\right)\\ & +\int \;\;\;\int dr_{0}dr\;\rho_{1}^{*}\left(r_{0}\right)\rho^{*}\left(r_{0}-r\right)\left\{\tilde{g}\left(r\right)\ln\tilde{g}\left(r\right)-\tilde{h}\left(r\right)-{\tilde{h}}^{2}\left(r\right)+e^{\tilde{h}\left(r\right)}-\tilde{g}\left(r\right)\right\}, \end{array}$$
where the third line of Equation (Equation 34) is obtained from the derivative of the logarithmic term in the fifth line of Equation (Equation 32) with respect to *m* using the Laplace expansion of $\left|S\right|$ along the first row as follows:(35)$$\begin{array}{cc} -\frac{1}{2}\frac{\partial }{\partial m}\ln\left|S(r)\right| & =-\frac{1}{2\left|S(r)\right|}\left(\frac{\partial \left|S(r)\right|}{\partial m}\right)\\ & =\frac{1}{2}\rho_{1}^{*}\left(r_{0}\right)\rho^{*}\left(r_{0}-r\right)\tilde{c}\left(r\right)\tilde{h}\left(r\right), \end{array}$$
where use has been made of the cofactor expansion in calculating $\partial \left|S(r)\right|/\partial m$.

It follows from Equation (Equation 34) that the stationary condition (Equation 33) becomes
(36)$$\begin{array}{c} {\left.\frac{\delta W(\tilde{G})}{\delta \tilde{h}}\right|}_{\tilde{h}={\tilde{h}}_{*}}\approx \frac{1}{N^{*}}\int dr_{0}\;\rho^{*}\left(r_{0}\right)\rho^{*}\left(r_{0}-r\right)\left\{{\tilde{c}}_{*}\left(r\right)+\ln{\tilde{g}}_{*}\left(r\right)-1-2{\tilde{h}}_{*}\left(r\right)+e^{{\tilde{h}}_{*}\left(r\right)}\right\}=0, \end{array}$$
where the subscript 1 has been dropped because of the indistinguishability of all replicas in the limits of m→1 and $\tilde{v}\left(r\right)\to 0$, $\delta \tilde{h}/\delta \rho^{*}$ and its inverse are ignored, and the first term on the rhs is an approximate form obtained from the third line of Equation (Equation 34) (see Appendix F for the detailed derivation). We can easily verify the equivalence between Equation (Equation 36) and the following closure: (37)$${\tilde{g}}_{*}\left(r\right)=e^{{\tilde{h}}_{*}\left(r\right)-{\tilde{c}}_{*}\left(r\right)+B\left(r\right)},$$
(38)$$B\left(r\right)={\tilde{g}}_{*}\left(r\right)-e^{{\tilde{h}}_{*}\left(r\right)},$$
which corresponds to the third result (*Result 3*), a new closure in the context of the liquid-state theory [62,63].

Two remarks on Equations (Equation 33), (Equation 37) and (38) are in order:Equation (Equation 33) is valid when a metastable state at ${\tilde{h}}_{*}\left(r\right)=\tilde{h}\left(r\right)$ is stable in the vanishing limit of the inter-replica interaction potential (i.e., $\tilde{v}\left(r\right)\to 0$); otherwise, transitions between basins occur in the FEL and the inter-replica correlations disappear, thereby amounting to ${\tilde{g}}_{*}\left(r\right)=1+{\tilde{h}}_{*}\left(r\right)=1$, the trivial solution to Equation (Equation 33). In other words, the new closure (Equation 37) applies to the metastable state defined by Equation (Equation 33).The bridge function B(r) given by Equation (38) is approximated by $B\left(r\right)=-{\tilde{h}}_{*}^{2}\left(r\right)/2$, which coincides with the main term of either the soft mean spherical approximation (MSA) or various approximations used for hard-sphere systems [63,85].

### 3.4. *Result 4*: Self-Consistent Equation for the Non-Ergodicity Parameter

In the fourth result (*Result 4*), we restrict ourselves to uniform systems in Fourier space. We introduce the non-ergodicity parameter f(k) by relating the inter-replica TCF ${\tilde{h}}_{*}\left(k\right)$ to the intra-replica structure factor $S\left(k\right)=1+\bar{\rho }h_{*}\left(k\right)$ [15,16,33,34,35,36,37,38]:(39)$$\begin{array}{c} f\left(k\right)=\frac{\bar{\rho }{\tilde{h}}_{*}\left(k\right)}{S(k)}. \end{array}$$
We need to find an approximation of the closure (Equation 37) that is available to obtain the self-consistent equation including terms up to quadratic order in the non-ergodicity parameter f(k). It is appropriate for this purpose to expand the rhs of the closure (Equation 37), providing
(40)$$\begin{array}{c} {\tilde{g}}_{*}\left(r\right)\approx {\tilde{g}}_{*}\left(r\right)-{\tilde{c}}_{*}\left(r\right)+B\left(r\right)+\frac{1}{2}{\left\{{\tilde{h}}_{*}\left(r\right)-{\tilde{c}}_{*}\left(r\right)\right\}}^{2}. \end{array}$$
Equation (Equation 40) reads in Fourier space
(41)$$\begin{array}{c} {\tilde{c}}_{*}\left(k\right)=\frac{1}{2}\int dq\;\left\{{\tilde{c}}_{*}\left(q\right)\;{\tilde{c}}_{*}\left(k-q\right)-{\tilde{c}}_{*}\left(q\right)\;{\tilde{h}}_{*}\left(k-q\right)-{\tilde{h}}_{*}\left(q\right)\;{\tilde{c}}_{*}\left(k-q\right)\right\}, \end{array}$$
when making the approximation of $B\left(r\right)\approx -h_{*}^{2}\left(r\right)/2$ as remarked after Equation (38). Meanwhile, the neglect of inhomogeneity (i.e., $\rho^{*}\left(r\right)=\bar{\rho }$) allows us to express the Fourier transform of the Ornstein-Zernike Equation (Equation 31) at m=1 as
(42)$$\begin{array}{c} {\tilde{c}}_{*}\left(k\right)=\frac{1}{\bar{\rho }S\left(k\right)}\left\{\frac{f(k)}{1-f(k)}\right\}, \end{array}$$
using the non-ergodicity parameter f(k) defined by Equation (Equation 39).

Combining Equations (Equation 39), (Equation 41) and (Equation 42), we obtain the self-consistent equation for f(k) (*Result 4*):(43)$$\begin{array}{c} \frac{f(k)}{1-f(k)}=\frac{S(k)}{2\bar{\rho }}\int dq\;M\left(k,q\right)\;S\left(q\right)\;S\left(k-q\right)\;f\left(q\right)\;f\left(k-q\right)+O\left[f^{3}\right], \end{array}$$
where the inverse of the intra-replica structure factor S(q) is related to the intra-replica DCF $c_{*}\left(q\right)$ as $1/S\left(q\right)=1-\bar{\rho }c_{*}\left(q\right)$ and the kernel M(k,q) is given by
(44)$$\begin{array}{cc} M(k,q) & =\frac{1}{S^{2}\left(q\right)\;S^{2}\left(k-q\right)}-\frac{1}{S^{2}\left(q\right)}-\frac{1}{S^{2}\left(k-q\right)}\\ & ={\left\{{\bar{\rho }}^{2}c_{*}\left(q\right)\;c_{*}\left(k-q\right)\right\}}^{2}+2{\bar{\rho }}^{2}c_{*}\left(q\right)\;c_{*}\left(k-q\right)\left\{\frac{1}{S(q)}+\frac{1}{S(k-q)}\right\}-1; \end{array}$$
see Appendix G for details. We can relate the product $c_{*}\left(q\right)\;c_{*}\left(k-q\right)$ in Equation (Equation 44) to the triplet DCF $c_{*}^{\left(3\right)}\left(q,k-q\right)$ by adopting the approximate form as follows:(45)$$\begin{array}{c} c_{*}^{\left(3\right)}\left(q,k-q\right)=\frac{c_{*}^{\left(3\right)}\left(0,0\right)}{{\left\{c_{*}\left(0\right)\right\}}^{2}}\;c_{*}\left(q\right)\;c_{*}\left(k-q\right), \end{array}$$
which is validated by the weighted density approximation or the closure-based density functional theory [76,77,78]. The expression (Equation 45) and the introduction of the negative factor, $\alpha ={\left\{c_{*}\left(0\right)\right\}}^{2}/c_{*}^{\left(3\right)}\left(0,0\right)<0$, transform Equation (Equation 44) into the following kernel (*Result 4*):(46)$$\begin{array}{c} M\left(k,q\right)={\left\{{\bar{\rho }}^{2}\alpha c_{*}^{\left(3\right)}\left(q,k-q\right)\right\}}^{2}+2{\bar{\rho }}^{2}\alpha c_{*}^{\left(3\right)}\left(q,k-q\right)\left\{\frac{1}{S(q)}+\frac{1}{S(k-q)}\right\}-1, \end{array}$$
where $\alpha ={\left\{c_{*}\left(0\right)\right\}}^{2}/c_{*}^{\left(3\right)}\left(0,0\right)$ and $c_{*}^{\left(3\right)}\left(q,k-q\right)$ denotes the triplet DCF [76,77,78]. It is noted that Equation (Equation 46) can be compared with the previous result from other static theories [37,56,57]: the systematic expansion methods lead to the appearance of the triplet DCF in the kernel [37,56,57], similar to Equation (Equation 46).

## 4. Derivation Process of *Result 2*

This section presents a scheme to obtain *Result 2* based on the strong-coupling perturbation theory (see Appendix B). To this end, we focus on how to perform the functional integrals over one-body and two-body potential fields appearing in Equations (Equation 85), (Equation 93), (Equation 102) and (Equation 105)–(Equation 107).

### 4.1. One-Body Potential Field (1): Evaluating Equation (Equation 105) in the Saddle-Point Approximation

We see from Equation (Equation 104) that the saddle-point equation ${\left.\delta H_{\mathrm{mf}}\left(\phi \right)/\delta \phi \right|}_{\phi_{a}=i\psi_{a}^{*}}=0$ in Equation (Equation 105) gives
(47)$$\begin{array}{c} {\left.\frac{\delta \beta H_{0}\left(c,\phi \right)}{\delta \phi_{a}}\right|}_{\phi_{a}=i\psi_{a}^{*}}=\left(\frac{\bar{\rho }}{\gamma }\right){\left.\frac{\delta U_{1}\left(\phi \right)}{\delta \phi_{a}}\right|}_{\phi_{a}=i\psi_{a}^{*}}. \end{array}$$
Substituting Equations (Equation 88) and (Equation 99) into Equation (Equation 47), we have
(48)$$\begin{array}{cc} \psi_{a}^{*}\left(r\right) & =\frac{c_{aa}\left(0\right)}{2}-e^{\beta \mu }\sum_{b=1}^{m}\int dr^{\prime }c_{ab}\left(r-r^{\prime }\right)e^{-\psi_{b}^{*}\left(r^{\prime }\right)}. \end{array}$$
We can verify that Equation (Equation 48) transforms to Equation (Equation 24) by setting $\rho_{a}^{*}\left(r\right)=e^{\beta \mu -\psi_{a}^{*}\left(r\right)}$.

Let $F_{\mathrm{mf}}\left(-k_{B}Tc,\rho^{*}\right)$ be the mean-field free energy defined by
(49)$$\begin{array}{c} F_{\mathrm{mf}}\left(-k_{B}Tc,\rho^{*}\right)=U\left(-k_{B}Tc,\rho^{*}\right)-TS^{\mathrm{id}}\left(\rho^{*}\right), \end{array}$$
where $U(v,\hat{\rho })$ has been defined in Equation (Equation 21) and $S^{\mathrm{id}}\left(\rho^{*}\right)$ denotes the sum of ideal gas entropy $S_{a}^{\mathrm{id}}\left(\rho_{a}^{*}\right)$ given by Equation (Equation 17): (50)$$\begin{array}{cc} -TS^{\mathrm{id}}\left(\rho^{*}\right) & =-T\sum_{a=1}^{m}S_{a}^{\mathrm{id}}\left(\rho_{a}^{*}\right)\\ & =k_{B}T\int dr_{0}\;\left[\;\rho_{1}^{*}\left(r_{0}\right)\left\{\ln\rho_{1}^{*}\left(r_{0}\right)-1-\beta \mu \right\}+\left(m-1\right)\rho^{*}\left(r_{0}\right)\left\{\ln\rho^{*}\left(r_{0}\right)-1-\beta \mu \right\}\right]. \end{array}$$
Plugging Equation (Equation 24) into Equations (Equation 49) and (Equation 50), we find
(51)$$\begin{array}{cc} \beta F_{\mathrm{mf}}\left(-k_{B}Tc,\rho^{*}\right) & =\frac{1}{2}\int \;\;\;\int drdr^{\prime }\rho^{*}{\left(r\right)}^{T}c\left(r-r^{\prime }\right)\rho^{*}\left(r^{\prime }\right)-\sum_{a=1}^{m}\int dr\;\rho_{a}^{*}\left(r\right)\\ & =\beta H_{\mathrm{mf}}\left(i\psi^{*}\right) \end{array}$$
(see also Appendix C for details of the last equality).

The quadratic terms due to fluctuations around the saddle-point path $i\psi^{*}$ are written as
(52)$$\begin{array}{cc} & \beta H_{\mathrm{mf}}\left(\varphi +i\psi^{*}\right)-\beta H_{\mathrm{mf}}\left(i\psi^{*}\right)\\ & \;\approx -\frac{1}{2}\int \;\;\;\int drdr^{\prime }\varphi^{T}\left(r\right)c^{-1}\left(r-r^{\prime }\right)\varphi \left(r^{\prime }\right)+\sum_{a=1}^{m}\frac{1}{2}\int dr\rho_{a}^{*}\left(r\right)\varphi_{a}^{2}\left(r\right)\\ & \;=-\frac{1}{2}\int \;\;\;\int drdr^{\prime }\varphi {\left(r\right)}^{T}h^{-1}\left(r-r^{\prime }\right)\varphi \left(r^{\prime }\right). \end{array}$$
In the last equality of Equation (Equation 52), use has been made of the following relation:(53)$$\begin{array}{c} h_{ab}^{-1}\left(r-r^{\prime }\right)=c_{ab}^{-1}\left(r-r^{\prime }\right)-\rho_{a}^{*}\left(r\right)\delta_{ab}\delta \left(r-r^{\prime }\right), \end{array}$$
which is equivalent to the inhomogeneous Ornstein-Zernike Equations (Equation 29) as confirmed in Appendix D. It is found from Equations (Equation 51) and (Equation 52) that the saddle-point approximation of Equation (Equation 105) yields
(54)$$\begin{array}{cc} e^{-\beta F(\nu =0)} & =\frac{1}{N_{c}}\;e^{-\beta F_{\mathrm{mf}}\left(-k_{B}Tc,\rho^{*}\right)}\int D\varphi \;e^{\frac{1}{2}\int \;\;\;\int drdr^{\prime }\varphi {\left(r\right)}^{T}h^{-1}\left(r-r^{\prime }\right)\varphi \left(r^{\prime }\right)}. \end{array}$$
Equations (Equation 87)–(Equation 89) further imply that Equation (Equation 54) is transformed into
(55)$$\begin{array}{cc} e^{-\beta F(\nu =0)} & =\frac{N_{h}}{N_{c}}\;e^{-\beta F_{\mathrm{mf}}\left(-k_{B}Tc,\rho^{*}\right)}\\ & =\frac{1}{N_{c}}\;e^{-\beta F_{\mathrm{mf}}\left(-k_{B}Tc,\rho^{*}\right)}\int D\varphi \;e^{-\beta H_{0}\left(h,\varphi \right)}. \end{array}$$
We will use the last line on the rhs of Equation (Equation 55) as a reference form in evaluating βF(ν)−βF(ν=0) given by Equation (Equation 106).

It follows from Equations (Equation 105) and (Equation 55) that
(56)$$\begin{array}{c} \beta \Phi_{-k_{B}Tc}^{\left(m\right)}=\beta F_{\mathrm{mf}}\left(-k_{B}Tc,\rho^{*}\right)-\ln\frac{N_{h}}{N_{c}}, \end{array}$$
where $N_{h}/N_{c}$ is related to the determinant of the matrix, $S=c^{-1}h$, as
(57)$$\begin{array}{c} \frac{N_{h}}{N_{c}}={\left\{\prod_{r,r^{\prime }}\left|S(r-r^{\prime })\right|\right\}}^{1/2}, \end{array}$$
and the matrix element of S is given by
(58)$$\begin{array}{cc} & S_{ac}\left(r-r^{\prime }\right)\equiv \sum_{b=1}^{m}\int dr\prime \prime \;c_{ab}^{-1}\left(r-r\prime \prime \right)h_{bc}\left(r\prime \prime -r^{\prime }\right)\\ & =\sum_{b=1}^{m}\int dr\prime \prime \left\{c_{ab}^{-1}\left(r-r\prime \prime \right)c_{bc}\left(r\prime \prime -r^{\prime }\right)+\sum_{d=1}^{m}\int du\;\rho_{d}^{*}\left(u\right)c_{ab}^{-1}\left(r-r\prime \prime \right)c_{bd}\left(r\prime \prime -u\right)h_{dc}\left(u-r^{\prime }\right)\right\}\\ & =\delta_{ac}\delta \left(r-r^{\prime }\right)+\sum_{d=1}^{m}\int du\;\rho_{d}^{*}\left(u\right)\delta_{ad}\delta \left(r-u\right)h_{dc}\left(u-r^{\prime }\right)\\ & =\delta_{ac}\delta \left(r-r^{\prime }\right)+\rho_{a}^{*}\left(r\right)h_{ac}\left(r-r^{\prime }\right)\\ & =\frac{\langle \rho_{a}\left(r\right)\rho_{c}\left(r^{\prime }\right)\rangle }{\rho_{c}^{*}\left(r^{\prime }\right)}\geq 0, \end{array}$$
ensuring that $\left|S|=\right|c^{-1}h|\geq 0$. Replacing r and $r^{\prime }$ by $r_{0}$ and $r_{0}-r$, respectively, in Equation (Equation 58), we have
(59)$$\begin{array}{c} -\ln\frac{N_{h}}{N_{c}}=-\frac{1}{2}\int \;\;\;\int dr_{0}dr\;\ln\left|S\right|, \end{array}$$
in agreement with the logarithmic term in Equation (Equation 32).

### 4.2. One-Body Potential Field (2): Perturbative Calculation of Equation (Equation 85)

Remembering that $\rho_{a}^{*}\left(r\right)=e^{\beta \mu -\psi_{a}^{*}\left(r\right)}$, the average term in Equation (Equation 106) becomes
(60)$$\begin{array}{cc} & {\left(\frac{\bar{\rho }}{\gamma }\right)}^{2}\;{\langle e^{\int dr\;\left\{i\phi_{1}\left(r\right){\hat{\rho }}_{1}^{\left(1\right)}\left(r\right)+i\phi_{a}\left(r\right){\hat{\rho }}_{a}^{\left(1\right)}\left(r\right)\right\}}\rangle }_{\phi }\\ & \;=\rho_{1}^{*}\left(r_{1,1}\right)\rho^{*}\left(r_{a,1}\right)\;{\langle e^{\int dr\;\left\{i\varphi_{1}\left(r\right){\hat{\rho }}_{1}^{\left(1\right)}\left(r\right)+i\varphi_{a}\left(r\right){\hat{\rho }}_{a}^{\left(1\right)}\left(r\right)\right\}}\rangle }_{\varphi }\\ & \;=\rho_{1}^{*}\left(r_{1,1}\right)\rho^{*}\left(r_{a,1}\right)\;e^{\tilde{h}\left(r_{1,1}-r_{a,1}\right)}, \end{array}$$
where the subscript φ denotes the following average:(61)$$\begin{array}{c} {\langle O\rangle }_{\varphi }=\frac{\int D\varphi \;O\;e^{-\beta H_{0}\left(h,\varphi \right)}}{\int D\varphi \;e^{-\beta H_{0}\left(h,\varphi \right)}}, \end{array}$$
according to Equation (Equation 55) (see Appendix E for the detailed derivation of Equation (Equation 60)).

It is noted that the one-particle densities, ${\hat{\rho }}_{1}^{\left(1\right)}\left(r\right)$ and ${\hat{\rho }}_{a}^{\left(1\right)}\left(r\right)$, of replicas 1 and *a* in Equation (Equation 60) represent the two-particle system as a mixture of two replicas. Accordingly, the last line on the rhs of Equation (Equation 60) reduces to $\rho^{*}\left(r_{a,1}\right)\rho_{1}^{*}\left(r_{1,1}\right)$ in the absence of inter-replica correlation between two particles of different replicas (i.e., $\tilde{h}\left(r_{a,1}-r_{1,1}\right)=0$) consistently with the following result for the sum of one-particle systems:(62)$$\begin{array}{c} {\langle \frac{\bar{\rho }}{\gamma }U_{1}\left(\phi \right)\rangle }_{\phi }=\sum_{a=1}^{m}\int dr_{a,1}\;\rho_{a}^{*}\left(r_{a,1}\right)\;{\langle e^{\int dr\;i\varphi_{a}\left(r\right){\hat{\rho }}_{a}^{\left(1\right)}\left(r\right)}\rangle }_{\varphi }=\sum_{a=1}^{m}\int dr_{a,1}\;\rho_{a}^{*}\left(r_{a,1}\right), \end{array}$$
where the above φ-averaging is applied to the one-particle term $U_{1}\left(\phi \right)$ given by Equation (Equation 99), or setting $O=\left(\bar{\rho }/\gamma \right)\;U_{1}\left(\varphi +i\psi^{*}\right)$ in Equation (Equation 61) because of $\phi =\varphi +i\psi^{*}$.

Combining Equations (Equation 83), (Equation 84), (Equation 93), (Equation 106) and (Equation 60), we obtain the additional contribution to $\beta \Phi_{v^{\prime }}^{\left(m\right)}$ given by Equations (Equation 78), (Equation 49), (Equation 56) and (Equation 59):(63)$$\begin{array}{cc} e^{-\beta \Omega_{v^{\prime }}^{\left(m\right)}\left(\tilde{G}\right)+\beta \Phi_{v^{\prime }}^{\left(m\right)}} & ={\langle \prod_{a=2}^{m}\Delta_{a}\left(\tilde{G},\rho \right)\rangle }_{c}\\ & =\int D^{\prime }\nu \;e^{-\sum_{a=2}^{m}\Gamma_{a}\left(\nu \right)}, \end{array}$$
and
(64)$$\begin{array}{c} \Gamma_{a}\left(\nu \right)=-\int \;\;\;\int drdr^{\prime }\;\rho_{1}^{*}\left(r\right)\rho^{*}\left(r^{\prime }\right)\left\{i\;\tilde{g}\left(r-r^{\prime }\right)\;\nu_{a}\left(r-r^{\prime }\right)+e^{\tilde{h}\left(r-r^{\prime }\right)}\;f\left(i\;\nu_{a}\right)\right\}; \end{array}$$
see Equation (Equation 83) for the definition of ${\langle O\rangle }_{c}$. The results from the strong-coupling perturbation method developed in Appendix B are summed up in Equations (Equation 63) and (Equation 64).

### 4.3. Two-Body Potential Field: Derivation of *Result 2*: Rearrangements in the Mean-Field Approximation of Equation (Equation 63)

There are two remaining steps toward obtaining Equation (Equation 32): the first step is to evaluate the ν-functional integral given by Equation (Equation 63), and the second step is to rearrange the interaction energy when adding the last two terms on the rhs of Equation (Equation 78) to $U(-k_{B}Tc,\rho^{*})$.

First, let us evaluate the ν–field integral given by Equation (Equation 63) in the mean-field approximation. Equations (Equation 63) and (Equation 64) provide the saddle-point equation as follows:(65)$$\begin{array}{c} {\left.\frac{\delta \Gamma_{a}\left(\nu \right)}{\delta \nu_{a}}\right|}_{\nu_{a}^{*}=iu}=0, \end{array}$$
giving
(66)$$\begin{array}{c} \tilde{g}\left(r\right)=e^{\tilde{h}\left(r\right)+u\left(r\right)}, \end{array}$$
similar to a closure in the liquid-state theory [62,63] though given correlation functions do not necessarily satisfy any closure, other than the Ornstein-Zernike equation. Substituting Equation (Equation 66) into Equation (Equation 64), we obtain
(67)$$\begin{array}{c} \Gamma_{a}\left(\nu^{*}\right)=\int \;\;\;\int dr_{0}dr\;\rho_{1}^{*}\left(r_{0}\right)\rho^{*}\left(r_{0}-r\right)\left[\;\tilde{g}\left(r\right)\left\{\ln\tilde{g}\left(r\right)-\tilde{h}\left(r\right)\right\}+e^{\tilde{h}\left(r\right)}-\tilde{g}\left(r\right)\right], \end{array}$$
or
(68)$$\begin{array}{cc} & \Omega_{v^{\prime }}^{\left(m\right)}\left(\tilde{G}\right)-\Phi_{v^{\prime }}^{\left(m\right)}\\ & \;=\left(m-1\right)\int \;\;\;\int dr_{0}dr\;\rho_{1}^{*}\left(r_{0}\right)\rho^{*}\left(r_{0}-r\right)\left[\;\tilde{g}\left(r\right)\left\{\ln\tilde{g}\left(r\right)-\tilde{h}\left(r\right)\right\}+e^{\tilde{h}\left(r\right)}-\tilde{g}\left(r\right)\right], \end{array}$$
due to Equations (Equation 63) and (Equation 64).

Next, we rewrite the interaction energy. Considering the expression (Equation 21) and the Ornstein-Zernike equation,
(69)$$\begin{array}{c} h_{aa}\left(0\right)=c_{aa}\left(0\right)+\sum_{b=1}^{m}\int dr^{\prime }\rho_{b}^{*}\left(r^{\prime }\right)h_{ab}\left(r-r^{\prime }\right)c_{ab}\left(r-r^{\prime }\right), \end{array}$$
we have
(70)$$\begin{array}{cc} U(-k_{B}Tc,\rho^{*}) & +\sum_{a=1}^{m}\sum_{b=1}^{m}\frac{1}{2}\int \;\;\;\int drdr^{\prime }\rho_{a}^{*}\left(r\right)\rho_{b}^{*}\left(r^{\prime }\right)g_{ab}\left(r-r^{\prime }\right)c_{ab}\left(r-r^{\prime }\right)\\ & =\frac{1}{2}\sum_{a=1}^{m}\int \;\;\;\int dr_{0}dr\;\rho_{a}^{*}\left(r_{0}\right)h_{aa}\left(r\right)\delta \left(r\right)\\ & =\frac{1}{2}\int \;\;\;\int dr_{0}dr\;\left\{\rho_{1}^{*}\left(r_{0}\right)h_{1}\left(r\right)\delta \left(r\right)+\left(m-1\right)\rho^{*}\left(r_{0}\right)h\left(r\right)\delta \left(r\right)\right\}. \end{array}$$
To clarify the difference between the bare interaction potentials of v and $v^{\prime }$, we also separate the intra-replica interaction term from the inter-replica one created by $\tilde{v}\left(r\right)=v_{a1}\left(r\right)=v_{1a}\left(r\right)$:(71)$$\begin{array}{cc} \sum_{a=1}^{m}\sum_{b=1}^{m}\frac{1}{2}\int \;\;\;\int dr_{0}dr & \rho_{a}^{*}\left(r\right)\rho_{b}^{*}\left(r^{\prime }\right)g_{ab}\left(r-r^{\prime }\right)v_{ab}\left(r-r^{\prime }\right)\\ = & \frac{1}{2}\int \;\;\;\int dr_{0}dr\;\left\{\rho_{1}^{*}\left(r_{0}\right)\rho_{1}^{*}\left(r_{0}-r\right)+\left(m-1\right)\rho^{*}\left(r_{0}\right)\rho^{*}\left(r_{0}-r\right)\right\}g\left(r\right)v\left(r\right)\\ & +\left(m-1\right)\int \;\;\;\int dr_{0}dr\;\rho_{1}^{*}\left(r_{0}\right)\rho^{*}\left(r_{0}-r\right)\tilde{g}\left(r\right)\tilde{v}\left(r\right). \end{array}$$
Combining Equations (Equation 49), (Equation 50), (Equation 56), (Equation 59), (Equation 68), (Equation 70) and (Equation 71), we obtain $\beta \Omega_{v^{\prime }}^{\left(m\right)}\left(\tilde{G}\right)$ expressed by Equation (Equation 32), namely *Result 2*.

## 5. Concluding Remarks

The generalized FP potential $W(\tilde{G})$ as a functional of given TCF $\tilde{h}\left(r\right)$ is similar to the original FP potential [65,66,67,68,69,70,71,72,73,74,75] in that both have constraints on inter-replica correlations. The difference is that the generalized FP potential adopts a local order parameter instead of a global order one, the overlap *Q* (see Equation (Equation 1)), used in the original FP potential V(Q). Upon reviewing the formulation of $W(\tilde{G})$ presented so far, we find two essentials for the field-theoretical achievements. The former lies in the variational method described in Section B.1, whereas Equation (Equation 83) represents the latter. The details follow:*Unconstrained grand potential mimicking inter-replica correlations*: At first, we consider a coupled *m*-replica system that reproduces a given distribution of the inter-replica TCF $\tilde{h}\left(r\right)$ without constraints. We tune the inter-replica interaction potential $\tilde{v}\left(r\right)$ to mimic the inter-replica correlations. From evaluating the free-energy functional without constraints in the Gaussian approximation, we obtain the same functional form as the random phase approximation (RPA) in terms of the liquid-state theory [62,63]; however, the density distribution is different. The variational method presented in Section B.1 justifies the input of the density distribution given by Equation (Equation 27), which converges to that of the Ramakrishnan–Yussouf density functional theory [61] in the limit of m→1 as demonstrated in Equation (Equation 28).*Evaluating the difference between the constrained and unconstrained grand potentials*: Next, we take the free-energy functional of the unconstrained system as a reference energy. Equation (Equation 83) indicates that the field-theoretical formulation focuses on the free energy difference between the constrained and unconstrained free-energy functionals. The strong-coupling expansion method developed in Section B.3 allows us to evaluate this difference in Section 4.2 and Section 4.3. Thus, we obtain Equation (Equation 68), the constraint-associated free energy difference as a functional of inter-replica TCF $\tilde{h}\left(r\right)$ and density distribution $\rho^{*}\left(r\right)$ determined by the Ramakrishnan-Yussouf theory [61].

Equation (Equation 68) reduces to the functional difference between the $\tilde{h}\left(r\right)$-dependent parts in the HNC and RPA approximations when substituting $e^{\tilde{h}\left(r\right)}-\tilde{g}\left(r\right)\approx {\tilde{h}}^{2}\left(r\right)/2$ into Equation (Equation 68). This agreement indicates consistency between the field-theoretical formalism in this paper and the Legendre-transform-based theory using the Morita-Hiroike functional [33,34,35,36,37,38,64].

Combination of Equations (Equation 23) and (Equation 34) gives the difference between the generalized FP potentials at zero and a finite value of the inter-replica TCFs as follows:(72)$$\begin{array}{cc} & W\left(\tilde{G}={\bar{\rho }}^{2}\left\{1+\tilde{h}\right\}\right)-W\left({\bar{\rho }}^{2}\right)\\ & \;=\frac{1}{2N^{*}}\int \;\;\;\int dr_{0}dr\;\rho^{*}\left(r_{0}\right)\rho^{*}\left(r_{0}-r\right)\tilde{c}\left(r\right)\tilde{h}\left(r\right)\\ & \;+\frac{1}{N^{*}}\int \;\;\;\int dr_{0}dr\;\rho^{*}\left(r_{0}\right)\rho^{*}\left(r_{0}-r\right)\left\{\tilde{g}\left(r\right)\ln\tilde{g}\left(r\right)-\tilde{h}\left(r\right)-{\tilde{h}}^{2}\left(r\right)+e^{\tilde{h}\left(r\right)}-\tilde{g}\left(r\right)\right\}. \end{array}$$
The potential difference in Equation (Equation 72) arises from the entropic cost of localizing the system in an arbitrary state. It is noted, however, that the closure given by Equations (Equation 37) and (38) applies only to Equation (Equation 72) in a metastable state characterized by ${\tilde{h}}_{*}\left(r\right)$, which is in contrast to the Morita-Hiroike functional covering only the inter-replica TCF that necessarily satisfies the conventional closure [62,63] of the liquid-state theory due to the Legendre-transform-based formalism. That is, the generalized FP potential expressed as Equation (Equation 72) has a characteristic inherited from the original FP theory, a Landau-type theory relevant to investigate the FEL. Furthermore, Equation (Equation 72) represents that our study provides the basis of Ginzburg–Landau-type theory [79] as an extension of Landau-type one: the generalized FP potential $W(\tilde{G})$ as a functional of local order parameter $\tilde{h}\left(r\right)$ is a natural extension of the FP potential V(Q) as a function of the global order parameter *Q*.

The stationary Equation (Equation 33) reveals that the new closure (Equation 37) corresponds to the mean-field equation of $W(\tilde{G})$ given by Equation (Equation 72). The closure (Equation 37) gives the self-consistent Equation (Equation 43), similar to the previous one that predicts a dynamical transition [37,56,57]; we need to quantitatively assess the validity of Equation (Equation 43) in terms of the dynamical transitions in simulation models. Equation (Equation 10) further suggests that we can go beyond the mean-field approximation as is the case with the Ginzburg-Landau-type theory: the greatest advantage of our replica field theory is to systematically improve the self-consistent equation by considering fluctuations of inter-replica correlation field $\tilde{h}\left(r\right)$. It remains to be addressed whether the modified self-consistent equation explains the blurring of dynamical transition into a crossover from relaxational to activated dynamics.

There is a caveat, turning our attention to the stability condition on $\tilde{h}\left(r\right)$: translational and rotational symmetries are broken in frozen phases. The violation becomes evident by expanding $\tilde{h}\left(r\right)$ around that at the uniform density as follows [58,59,60]:(73)$$\begin{array}{c} \tilde{h}\left(r-r^{\prime };\rho^{*}\left(r\right)\right)=\tilde{h}\left(r-r^{\prime };\bar{\rho }\right)+{\left.\int dr^{\prime }\frac{\delta \tilde{h}\left(r-r^{\prime }\right)}{\delta \rho }\right|}_{\rho =\bar{\rho }}\left\{\rho^{*}\left(r^{\prime }\right)-\bar{\rho }\right\}+\cdots . \end{array}$$
We also have a non-perturbative approach to avoid the difficulty using a globally weighted density ${\bar{\rho }}_{\mathrm{WD}}$ in the inter-replica TCF: $\tilde{h}\left(r-r^{\prime };\rho^{*}\left(r\right)\right)=\tilde{h}\left(r-r^{\prime };{\bar{\rho }}_{\mathrm{WD}}\right)$, according to the modified weighted density functional approximation [43,48]. Therefore, the functional derivative in Equation (Equation 36), or the new closure (Equation 37), holds approximately when either neglecting the second and higher-order terms in Equation (Equation 73) or finding ${\bar{\rho }}_{\mathrm{WD}}$.

The new closure (Equation 37) in a metastable state provides the self-consistent Equation (Equation 43) for the non-ergodicity parameter f(k). The present field theory has demonstrated the necessity to consider higher-order contributions in the perturbative treatment for obtaining the self-consistent Equation (Equation 43) with a kernel containing the triplet DCF [76,77,78]: we obtain Equation (Equation 43) by adopting the approximate bridge function $B\left(r\right)=-{\tilde{h}}^{2}\left(r\right)/2$ beyond the RHNC approximation of B(r)=0. For comparison, we would like to mention two previous replica approaches to provide the triplet DCF in the self-consistent equation [37,56,57]. The first approach considers the perturbative contribution to the replicated HNC functional along the liquid-state theory [37]. The Legendre-transform-based method allows us to calculate the third order in $\tilde{h}\left(r\right)$ concerning the Morita-Hiroike functional. Meanwhile, the second method considers the third-order term in density difference $\rho^{*}\left(r\right)-\bar{\rho }$ by taking the Ramakrishnan-Yussouf functional of the DFT as a reference form [56,57]. Consequently, both perturbation methods amount to having the triplet DCF in the kernel of the self-consistent equation. This agreement implies the equivalence between the replicated HNC and Ramakrishnan–Yussouf approximations, consistent with the conventional results of the liquid-state theory [62].

Our scheme bears similarity to the Legendre-transform-based theory [33,34,35,36,37,38] rather than the DFT [53,54,55,56,57]. However, more elaborate input from the DFT [58,59,60] is also to be investigated, which is particularly necessary to investigate the glass transition in thin polymer films [93,94]; for example, we can improve the Ramakrishnan–Yussouf approximation by performing a variational evaluation beyond the Gaussian approximation (see Section B.1). Furthermore, our replica theory has two additional features arising from the field-theoretical treatment of the inter-replica TCF $\tilde{h}\left(r\right)$ and the associated two-body interaction potential $i\nu_{a}\left(r\right)$. First, we can systematically consider fluctuations around the mean-field potential field, $\nu_{a}^{*}\left(r\right)=iu\left(r\right)$, given by Equation (Equation 65), which is the same relation as that of the Legendre-transform-based method [33,34,35,36,37,38,64,86]. Second, we can develop the replica field theory to include TCF fluctuations around the metastable field ${\tilde{h}}_{*}\left(r\right)$ as described above. Thus, the present field-theoretical formalism opens up promising avenues to advance studies on the dynamical heterogeneity in terms of the correlation function of TCF fluctuations (i.e., the so-called four-point correlation function [7,8,9,10]) as well as the FEL that includes fluctuations around a metastable state. 

## Figures and Tables

**Figure 1 entropy-26-00241-f001:**
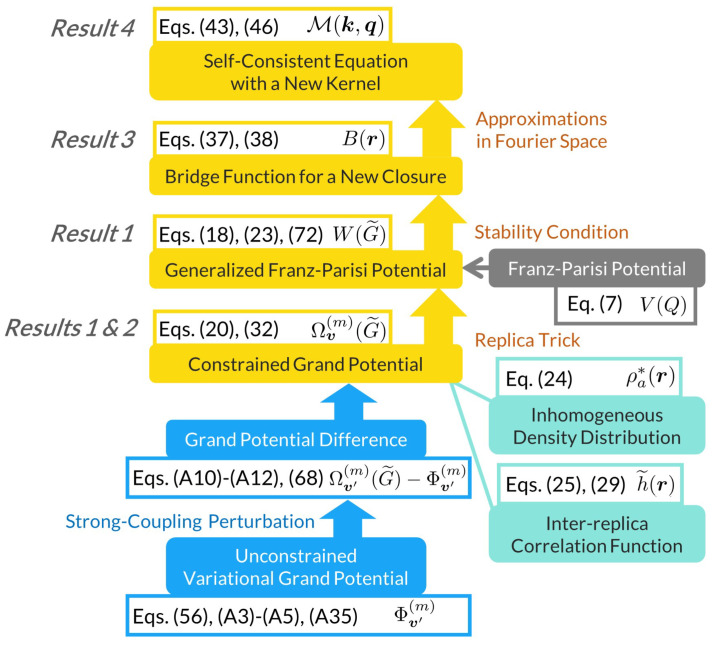
A schematic summary of the main results colored orange. In addition, functional variables are colored green, and underlying potentials blue or gray.

## Data Availability

No new data were created or analyzed in this study. Data sharing is not applicable to this article.

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
