# Peer review of "Replica Field Theory for a Generalized Franz–Parisi Potential of Inhomogeneous Glassy Systems: New Closure and the Associated Self-Consistent Equation"

_entropy, 2024, doi:10.3390/e26030241_

Round 1

Reviewer 1 Report

Comments and Suggestions for Authors

In the paper the author presents a generalization of the Franz-Parisi potential in which instead of constraining the the "reference" and the slaved configuration using a global order parameter such as the overlap, one fixes local order parameter such as the correlation function. Similarly to previous results, the generalization of the Franz-Parisi potential can be calculated computing the free energy of a system of m coupled replicas. The stationarity condition of the new Franz-Parisi functional allows the author to derive a new self-consistent equation involving triplet direct correlation function. 

Overall I feel that only the abstract and introduction are well written. I am not judging the result of the paper, that seem sound and appropriate for the journal, but the way they are presented and derived. I already had problems in following simple steps appearing in the first pages, due to a possible confusing notation used or a lack of clear explanations. With the current notation or way of presentation an average/non-familiar reader is not able to understand what the author is doing. 

Some examples:

- in the first line of eq 6, after the equal sign, where is the dependence on C_1? 

- Eq 7 is not effectively used in eq 6, at least for one sees in the first line of equation 8. If I am wrong, please explain better, or simply avoid equation 7, by directly defining W as in the first line of 8.

- Eq. 8: from the first to the second line a simple identities with delta functions are inserted. As a result U_a can be seen as a function of non-hat densities i.e. rho_1 and rho_a. But from the third to the fourth line the integration over \rho_1 is done again, so it seems that the author is doing a trivial step in reverse, with a replacement of \hat \rho_1 with \rho_1, which is non-sense. The explanation in line 144-147 is not sufficiently clear.

Moreover I feel that the way of exposition can be greatly improved. Indeed at the present stage the exposition of the result is greatly not linear, and in order to understand how a result is derived sometimes one has to extensively search through the paper. Figure 1 does not help at all. 

It is like reading a proof sketch (this is the idea for which Section 3 was written, I guess), but where every missing step is written somewhere in the document (and sometimes it is not even indicated or referenced where to find it). For example after stating result 2 (equation 24), one should correctly reference where to find the derivation, and, possibly, explicitly state how it is derived. 

Sometimes some of the result are used along the way (see for example the start of section 4.2) before even showing how to derive it independently! Only "result 1" is derived in section 3, (even if this is an extension of relation found by Monasson to non-local order parameters). One suggestion that I can do to help the author to improve the paper is to correctly reference where each result is carefully derived, moving more stuff in the appendix if possible. For example, I think that a derivation of result 2 could be moved in the appendix.

Finally, a reader starting to read section 4 would not understand if it is totally disconnected from section 3 (e.g. new grandpotential are introduced in a very brief way). Therefore I suggest to extend the introduction to section 4, describing more in the detail what is the goal of the section and how it compares with section 3 (for example better describing what is the starting quantity of interest, how it relates to constrained grand potential of m replicas \Omega and how and where the result of section 3 will be obtained along the way).

Minor points:

- in line 137 the author can explicitly present a possible function \eta(r).

- Right after equation (2): it could be better if the author can anticipate a possible definition of the interaction energy just to better show the dependence on \hat rho. Only appears in eq 13 for the first time. This will better clarify some of the confusion that an average reader could have in reading the rest of the paper. 

- in eq (3) the beta factor on the lhs and the 1/N on the rhs should be removed in order to interpret V as a free energy, as I think the author wants to. 

Reviewer 2 Report

Comments and Suggestions for Authors

This paper develops a generalized Franz-Parisi potential to study inhomogeneous glassy systems. The research of glass is very important, and the approach adopted is closely related to some of the most popular ones in the literature. It can be a worthwhile study. The calculations appear sounded, despite the use of many approximations which may be unavoidable. Nevertheless, any possible eventual relevance of the work depends on whether experiments or simulations can be better explained, on which the paper discusses insufficiently. I suggest the author to consider the followings:

(1)    This is a long paper with many technical results. I think the work can generate more impacts by including only the main results in this paper but adding sections on applications to realistic or close-to-realistic systems. Usefulness of the approach should hopefully be demonstrated by the applications. However, the present form may still be acceptable if the author at least discusses in detail and convincingly what can be compared directly with experiments or simulations and why improved agreements may be expected.

(2)    An important example of inhomogeneous glassy systems is glassy film, which is not discussed at all. Can the theory be applied to films? How? Or does the author have other systems in mind?

(3)    The author points out insightfully in the introduction that handling activated dynamics is a main challenge. Can the present results contribute?

(4)    In addition, going beyond the mean-field is also very important. Can the present results alter the mean-field predictions qualitatively to provide a more “stretched out” transition more consistent with experiments?

(5)    The introduction only reviewed replica and MCT methods. Recent progress on other approaches such as facilitation, elastic interactions and lattice simulations should also be mentioned, especially because of recently revived interest in facilitation. If possible, the merits, deficiencies and possible remedies of these approaches may be briefly discussed.

(6)    Eq. (6) is the most important equation but is only briefly explained. Quantities involved should be explicitly defined and explained.

(7)    How does tilde-G(r,r’) changes around the transition? How are each replica related in the glass phase? These might have been explained in passing but more details would be helpful. Some sketches of the function, if possible, would be nice.

(8)    In Eq. (17), when supposing a radial distribution function tilde-g(r), is the inhomogeneous assumption invalidated or limited?

(9)    The abstract mentions “both benefits”. I cannot be sure from the preceding text what they are.

Round 2

Reviewer 1 Report

Comments and Suggestions for Authors

The author has answered the raised criticisms and modified the paper accordingly.

I recommend publication of the manuscript in the present form.

Reviewer 2 Report

Comments and Suggestions for Authors

My concerns are addressed. I recommend publication in its present form.